# Practical Rubber Pre-Treatment Approch for Concrete Use—An Experimental Study

Rajeev Roychand [1,*], Rebecca J. Gravina [1,*], Yan Zhuge [2], Xing Ma [2], Julie E. Mills [2] and Osama Youssf [2,3]

1    School of Engineering, RMIT University, Melbourne, VIC 3000, Australia
2    UniSA STEM, University of South Australia, Adelaide, SA 5095, Australia; yan.zhuge@unisa.edu.au (Y.Z.);
     xing.ma@unisa.edu.au (X.M.); julie.mills@unisa.edu.au (J.E.M.); osama.youssf@unisa.edu.au (O.Y.)
3    Structural Engineering Department, Mansoura University, Mansoura 35516, Egypt
*    Correspondence: rajeev.roychand@rmit.edu.au (R.R.); rebecca.gravina@rmit.edu.au (R.J.G.)

**Abstract:** There is a lot of ongoing active research all over the world looking for various applications of used tyre rubber, to increase its utilisation rate. One of the common research applications is to incorporate rubber into concrete as a partial replacement for conventional aggregates. However, due to its poor bonding performance with cement paste, the utilisation of rubber in concrete has been hindered to date. A cost-effective and time-saving rubber pre-treatment method is of great interest, especially for the concrete industry. Out of all the various pre-treatment methods, soaking rubber particles in water is the most cost-effective and least complex method. In addition, sodium sulphate accelerates the hydration reaction of the cement composites. This study looks at the effect of soaking crumb rubber in tap water for short (2 h) and long (24 h) durations, and the optimised duration was then compared with soaking the crumb rubber in a 5% concentration of sodium sulphate solution. Compressive strength, bond behaviour, and rubber/cement interfacial transition zone (ITZ) were investigated using X-ray diffraction (XRD) and scanning electron microscopy (SEM) analysis. The results demonstrate that a soaking duration of 2 h provides much better performance in both the strength and bond properties compared to 24-h soaking. A further improvement in the 7-day strength was achieved with the rubber soaked in 5% sodium sulphate solution for 2 h, providing a more practical and economical rubber pre-treatment method for concrete industry use.

**Keywords:** crumb rubber; rubber mortar; rubber pre-treatment; XRF; XRD; SEM; ITZ

## 1. Introduction

With extensive research spanning the last few decades, researchers have found various applications for increasing the utilisation rate of waste tyres, such as (i) tyre re-treading; (ii) rubber-moulded products like flooring materials and shock absorbing playground mats; (iii) tyre pyrolysis to produce carbon black and oil/gas that can be used as a fuel; (iv) geotechnical applications like sub-grade fill in roads and embankments [1–4]; (v) rubber modified asphalt pavements [5]; and (vi) in concrete, as a replacement for aggregates [6–9]. Among all of these applications of tyre recycling, using rubber as a replacement for fine aggregates has been researched extensively in the last few years. The high cost of transporting sand and its scarcity have also added increased motivation to look for more alternatives [10–12]. Concrete incorporating rubber is called crumb rubber concrete (CRC). Using recycled tyre rubber waste in concrete can have many environmental benefits and can help save natural sand [13,14].

A significant body of small-scale or laboratory research on CRC has been carried out to date, e.g., [15–18]. CRC can show improved dynamic properties compared to conventional concrete, such as; toughness, ductility, impact resistance, and damping ratio [6,19]. However, it has construction use limitations due to its relatively low compressive strength, tensile strength, and modulus of elasticity [20]. This is due to the poor rubber bond performance with the cement paste [21,22] and its inherent soft material property [23,24].

However, with the advancement in research a range of rubber pre-treatment methods have been developed that not only improve the bond behaviour with the hardened cement paste, but also contribute towards improving the mechanical properties of crumb rubber concrete/mortar [25].

Mohammadi and Khabaz [26] studied the effect of soaking crumb rubber in water for 24 h on crumb rubber concrete at the fine aggregate replacement levels of 10, 20, 30, and 40%. Although their study did not make a comparison of the treated and untreated rubber concrete, their strength results for the treated rubber concrete showed a reduction of 14.1, 29.7, 51, and 63.7% in the compressive strength results at the corresponding replacement levels. In a recent study, Youssf et al. [14] investigated the comparative effect of treating rubber particles with different chemical solutions of NaOH, KMnO4+NaHSO$_4$, H$_2$O$_2$, CaCl$_2$, and H$_2$SO$_4$ on the mechanical properties of rubber concrete, containing a rubber content of 20% by volume of sand. They observed that treating the rubber particles with H$_2$O$_2$, H$_2$SO$_4$, and a combination of KMnO4 and NaHSO$_4$ did not bring about any considerable changes in the compressive strength results of the treated rubber concrete in comparison to the untreated rubber concrete. However, treating the rubber particles with NaOH and CaCl2 solutions brought about a similar but small strength improvement of ~7% compared to the untreated rubber concrete. A similar small improvement in the compressive strength results of NaOH treated rubber concrete was reported by Najim and Hall [27]. However, another study by Youssf et al. [14] found that NaOH rubber treatment for more than 0.5 h had an adverse effect on the compressive strength of the treated rubber concrete.

A comprehensive review on the effect of various rubber treatment methods on the mechanical properties of crumb rubber concrete was carried out by Roychand et al. [25], which covered research work published in last 30 years. The various rubber treatment methods they reported were: 24-h water soaking [28]; NaOH [29–32]; Silane coupling agent [33]; polyvinyl alcohol [34]; partial oxidation [35]; organic sulphur compounds [36]; UV [37] and gamma [38] radiations; solvents like methanol, ethanol, and acetone [39]; KMnO$_4$ and NaHSO$_3$ [40]; heat treatment [41]; acid treatments with H$_2$SO$_4$ [42]; HCl [43]; HNO$_3$ [44]; CH$_3$COOH [42]; Ca(OH)$_2$ [42]; CaCl$_2$ [29]; H$_2$O$_2$ [29]; and (xxi) CS$_2$ [45]. They made the following conclusions:

- out of the various methods that provide the least amount of preparations for the rubber treatment, the solvent treatment method using acetone provided the highest improvement in the compressive strength
- among the other rubber treatment methods that come with high treatment complexity, partial oxidation of the rubber particles at 250 °C provided the highest improvement in the compressive strength of the rubber concrete.

Although various rubber pre-treatment methods have shown improvements in the bond performance and mechanical properties of crumb rubber concrete, the time-consuming complex processes involved in the rubber treatment processes make them un-attractive and impractical for the concrete industry. Therefore, finding an economical and practical method of producing commercially viable CRC is of great interest. Water soaking of crumb rubber is the most cost-effective method as it does not require any additional chemicals or equipment and involves the least complexity in terms of processing. Previous studies that investigated rubber water pre-treatment have only looked at soaking the rubber particles for 24 h [28]. This relatively long time can be a hindrance for implementation in the fast-paced construction industry. Therefore, to address this research gap, this study investigated the effect of soaking crumb rubber in tap water for a short 2-h duration in comparison to the long 24-h duration investigated in the previous studies. The optimum duration of the rubber soaking period was further explored to identify the effect of soaking crumb rubber in 5% concentration of sodium sulphate solution to identify its potential benefits on the strength properties of the rubber mortar. Compressive strength, bond behaviour, and rubber/cement interfacial transition zone (ITZ) were investigated using X-ray diffraction

(XRD) and scanning electron microscopy (SEM) analysis of rubber cement mortar made by replacing 20% of sand volume with crumb rubber.

## 2. Experimental Programme
### 2.1. Materials and Methods

The binder material used in this experimental study was general blended (Type GB) cement. The chemical and material compositions of GB cement are shown in Tables 1 and 2, respectively. The specific gravity of 3.08 river sand with 5 mm size was used. The unit weight and specific gravity of the sand were 1420 kg/m$^3$ and 2.63, respectively. Crumb rubber of particles with a size range of 1.18–2.36 mm, sourced from "Tyrecycle pty ltd" (Tyre recycling company), was used as partial volume replacement for sand. The unit weight and specific gravity of rubber were 530 kg/m$^3$ and 0.97, respectively. A sieve analysis of the rubber and sand used is shown in Figure 1. Polycarboxylic-ether based superplasticizer with a specific gravity of 1.085 was used in this study.

**Table 1.** Chemical composition of GB cement.

| Material | SiO$_2$ | Al$_2$O$_3$ | CaO | MgO | Fe$_2$O$_3$ | SO$_3$ | Na$_2$O | K$_2$O | LOI |
|---|---|---|---|---|---|---|---|---|---|
| GBC | 26.4% | 8.7% | 53.4% | 3.8% | 1.9% | 2.4% | 0.24% | 0.35% | 1.9% |

**Table 2.** Material composition of GB cement.

| Material | Clinker | Slag | CaSO$_4$ | Limestone |
|---|---|---|---|---|
| GBC | 44% | 50% | 4% | 2% |

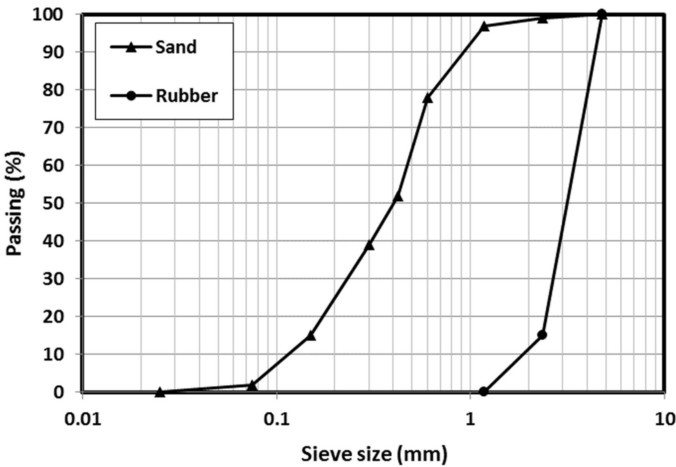

**Figure 1.** Sieve analysis of sand and rubber used.

The mixes were designed in accordance with AS 1012.2 [46]. Typically, based on our previous research [9,18,25,41] and the existing literature [47–49] published on this topic, the compressive strength of crumb rubber concrete decreases with the increase in rubber content. Therefore, a judicious decision has to be made to balance the environmental benefits of increasing the uptake of waste tyre rubber and the acceptable level of reduction in the compressive strength of rubber concrete. Increasing the rubber content higher than 20% brings about a significant reduction in compressive strength of the blended concrete. Therefore, a 20% replacement level was chosen for this experimental work which is considered the acceptable level of replacement in the majority of the published research. The mixing procedure for both control mortar and untreated rubber mortar mixes was as follows: mix dry sand, rubber (if any), and cement for 1 min; add water and admixtures, and then mix for 2 min. Final high-speed mixing was then followed for 1 min. For all other

mixes containing water-soaked rubber, the excess water was strained out using a fine mesh strainer. The moist crumb rubber was then mixed with cement for 1 min for an effective coating of cement particles over the crumb rubber. Then sand was added and mixed for another 1 min followed by 2 min mixing after the addition of water and superplasticizer. The fresh mortar was poured into 50 mm cubic moulds for compressive strength tests at different concrete ages (three cubes per test). All specimens were de-moulded after 24 h and labelled for the various tests. The strength test of the mortar specimens was conducted using 0.36 MPa/s loading rate according to ASTM C109 [50]. Table 3 shows the proportions of mixes used in this study.

**Table 3.** Mortar mix design.

| Mix | Method of Crumb Rubber Treatment | GBC (grams) | Sand (grams) | Rubber Content (by Volume of Sand) | Rubber (grams) | Water (mL) | SP (mL) |
| --- | --- | --- | --- | --- | --- | --- | --- |
| C | Control (No rubber) | 1000 | 2750 | 0% | 0 | 350 | 8 |
| M1 | Raw crumb rubber | 1000 | 2200 | 20% | 204 | 350 | 10 |
| M2 | Crumb rubber soaked in water for 2 h | 1000 | 2200 | 20% | 204 | 350 | 8 |
| M3 | Crumb rubber soaked in water for 24 h | 1000 | 2200 | 20% | 204 | 350 | 8 |
| M4 | Crumb rubber soaked in 5% sodium Sulphate solution for 2 h | 1000 | 2200 | 20% | 204 | 350 | 8 |

GBC = General blended cement, SP = Superplasticizer, Specific gravity of rubber = 0.97, Specific gravity of sand = 2.62.

### 2.2. Water Absorption/Adsorption by Crumb Rubber

The crumb rubber particles have a rough and vesicular surface that can trap water molecules in their vesicular cavities. In addition, they contain a large amount of carbon black, which has a good adsorption capacity [51]. In order to better analyse the results in this study, a water absorption/adsorption test for rubber particles was carried out. The particles were soaked in water for two separate durations, namely 2 h and 24 h each. The rubber particles were pushed into the water and left soaking for the different durations of soaking. Since the density of air is 1.225 kg/m$^3$, which is significantly lower than that of water, i.e., 1000 kg/m$^3$, once the particles are submerged in water, they initially displace the adsorbed low-density air molecules and take their place. Over time, the air trapped in the vesicular cavities is slowly displaced by the water molecules, increasing the amount of absorbed/adsorbed water in the rubber particles. This is evident from the different percentages of absorbed/adsorbed water between the 2 and 24 h soaking durations, as shown in Table 4. Figure 2 shows the freshly dropped rubber particles floating on water (left) and the same rubber particles submerged in water after they displace the low-density air molecules with high-density water molecules (right), increasing the density of the rubber particles, thereby making them submerge in the soaking water. The crumb rubber particles were then taken out of the soaking container after the designed soaking periods and the excessive water was strained off with the help of a 300 μm fine mesh strainer. The strained rubber was then weighed and dried for 24 h in an oven with temperature of 40 °C. After drying, the rubber was weighed again to check the absorption of water by the crumb rubber particles, which is expressed as the ratio between rubber weight increase due to water absorption/adsorption and the dry rubber weight. Table 4 presents the water absorption/adsorption by the rubber particles after different soaking periods. As can be observed in the table, the water absorbed/adsorbed by rubber after 24 h was only 24% higher than that after 2 h. This indicated that the rate of rubber water absorption/adsorption did not linearly increase with the soaking time, and the longer soaking time did not add much to the rubber water pre-treatment benefits.

**Table 4.** Crumb rubber water absorption/adsorption at 2 h and 24 h.

| Material | Percentage of Water Absorption | |
| --- | --- | --- |
| | 2 h Water Soaking | 24 h Water Soaking |
| Crumb Rubber | 4.00% | 4.95% |

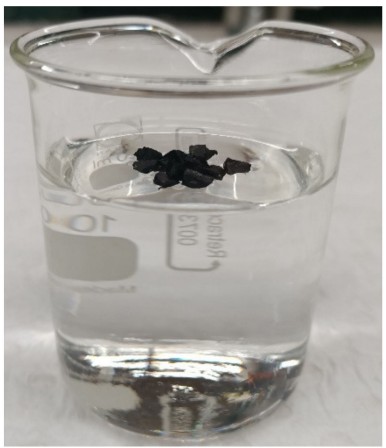
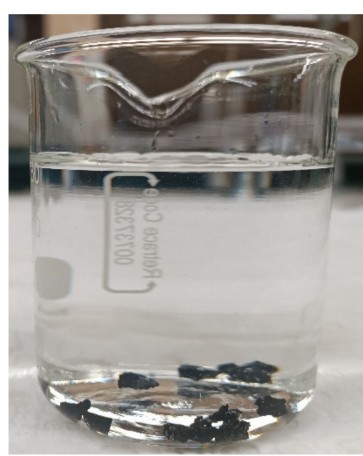

**Figure 2.** Pictures showing the crumb rubber freshly dropped onto the surface of water (**left**) and the rubber submerged in water after it displaces the air absorbed/adsorbed by the rubber particles with the water molecules (**right**).

### 2.3. SEM-EDS of Crumb Rubber

SEM images were taken on carbon coated raw crumb rubber and the crumb rubber soaked in 5% sodium sulphate solution for 2 h using FEI Quanta 200 SEM at 2000× magnification to understand the surface morphology and the effect of soaking crumb rubber particles in sodium sulphate solution. Figure 3 presents the SEM images of Figure 3a–c raw rubber and (d) rubber soaked in 5% sodium sulphate solution for 2 h. The SEM image of rubber particles soaked in the sodium sulphate solution shows a deposition of a large number of sodium sulphate particles. This further reinforces the XRD observation that the sodium sulphate solution does not react with any chemical component of the raw rubber and just gets deposited on the surface of the rubber particles. EDS analysis was undertaken on the untreated and treated rubber particles soaked in 5% sodium sulphate solution for 2 h and then dried at 40 °C temperature. Figure 3e show the EDS analysis of untreated crumb rubber, and Figure 3f,g shows the EDS analysis of the sodium sulphate particles deposited on the rubber surface. The EDS analysis of white particles deposited on the surface of the rubber particles soaked in sodium sulphate solution clearly show Na, S, and O elements, indicating the deposition of sodium sulphate particles. However, there was the presence of the element Carbon in the EDS analysis, which was most likely because of the electron beam spilling on the rubber surface, generating X-rays from the carbon present in the crumb rubber.

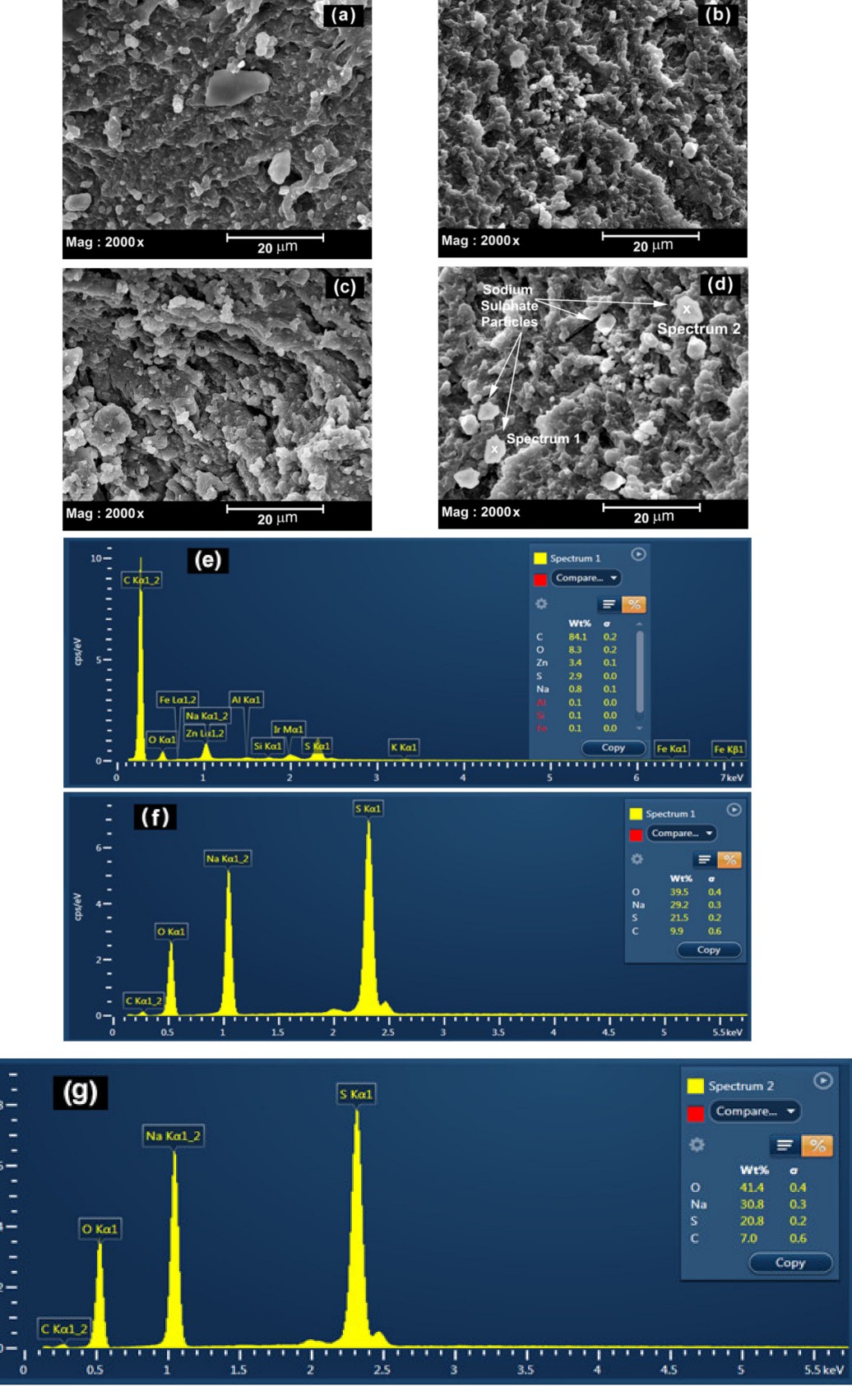

**Figure 3.** SEM images of (**a**–**c**) raw rubber, (**d**) rubber soaked in 5% sodium sulphate solution for 2 h, (**e**) EDS analysis of untreated crumb rubber, (**f**) Spectrum 1, and (**g**) Spectrum 2 of the EDS analysis of the rubber particles soaked in 5% sodium sulphate solution.

### 2.4. Carbon, Hydrogen, Nitrogen, and Sulphur (CHNS) Analysis of Crumb Rubber

CHNS analysis was undertaken on the rubber particles to identify the Carbon, Hydrogen, Nitrogen, and Sulphur content in the rubber particles that, combined with the EDS analysis, helped in the identification of the X-ray diffraction peaks and amorphous contents. Table 5 shows the CHNS analysis of the rubber particles

**Table 5.** CHNS analysis of the crumb rubber particles.

| Carbon (C) | Hydrogen (H) | Nitrogen (N) | Sulphur (S) |
|---|---|---|---|
| 83.34 % ± 0.32 | 4.37 % ± 1.94 | 0.53 % ± 0.01 | 2.09 % ± 0.16 |

### 2.5. XRD of Raw Material

X-ray diffraction was carried out on the following materials: GBC, sodium sulphate, raw crumb rubber, and crumb rubber soaked in 5% sodium sulphate solution for 2 h. The XRD equipment and settings were as below:

- XRD equipment: Bruker AXS-D4-Endeavour
- XRD detector: Lynxeye linear-strip detector
- X-ray source: Cu-K$\alpha$ radiation
- Operating current and voltage: 40 mA current and 40 kV voltage
- Testing range: 5° to 70° 2-theta
- Step size: 0.01° 2-theta

The XRD of crumb rubber, soaked in 5% sodium sulphate solution for 2 h, was conducted to identify whether the sodium sulphate solution reacted with the rubber particles and changed its properties or was simply absorbed/adsorbed by the rubber particles. Figure 4 shows the XRD diffractograms of GBC, sodium sulphate, raw crumb rubber, and crumb rubber soaked in 5% sodium sulphate solution for 2 h, along with their corresponding known crystalline and amorphous phases. The XRD diffractograms do not show any chemical change in the rubber particles with 2 h soaking in 5% sodium sulphate solution, indicating that the sulphate solution is only absorbed/adsorbed by the rubber particles and does not react with the rubber particles.

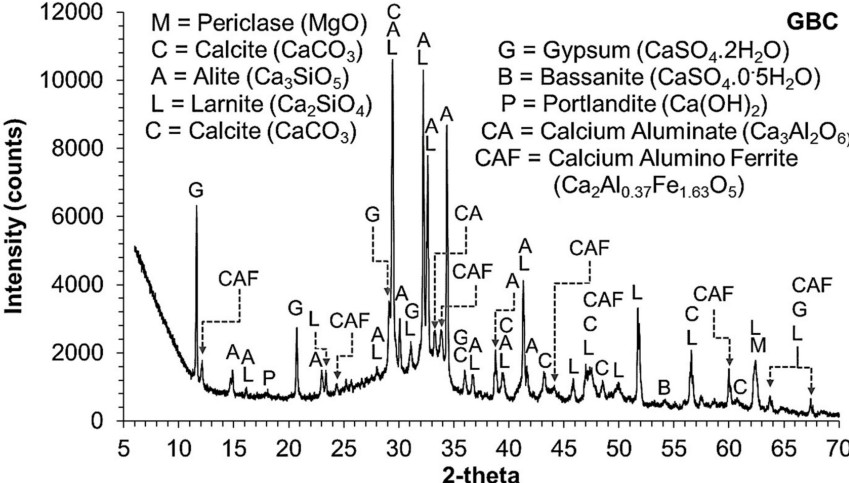

**Figure 4.** *Cont.*

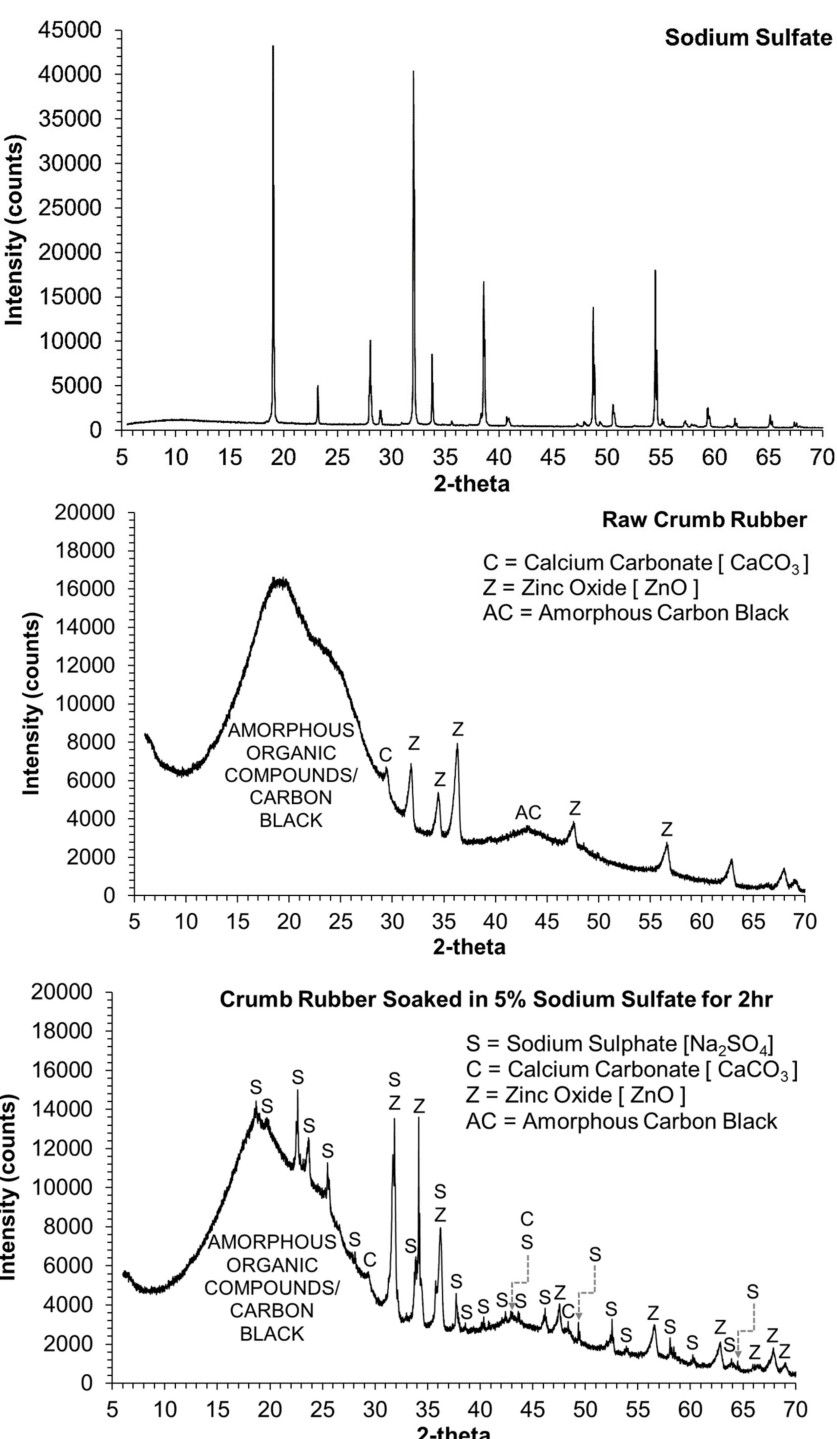

**Figure 4.** XRD diffractograms of: GBC, sodium sulphate, raw crumb rubber, and crumb rubber soaked in 5% sodium sulphate solution for 2 h.

### 2.6. XRD of Hardened Cement Paste of GBC and Mix M4

X-ray diffraction was carried out on the hardened cement pastes of GBC and mix M4 to ascertain the chemical changes occurring in the cement matrix due to the addition of sodium sulphate. The hardened cement pastes of (i) GBC containing normal tap water and (ii) mix M4 containing 5% sodium sulphate solution were cured for 28 days before conducting XRD. The samples were then micronized in a ring mill and sieved through a 45 μm sieve. To stop further hydration and to remove the physically bound water, the

solvent exchange method was adopted using acetone [52–54]. The instrument settings for conducting XRD were kept the same as described in Section 2.3.

### 2.7. SEM-EDS of Mortar Samples

A small slice was taken out of the internal core of the hardened mortar sample, embedded in an epoxy resin in 25 mm diameter Teflon moulds, and cured for 24 h. After 24 h the sample embedded in the epoxy was ground with 600 to 1200 grit silicon carbide grinding papers followed by further polishing of the samples with 9, 3, and 1 micron diamond suspensions. The polished specimens were mounted on steel stubs and gold coated for SEM imaging. The SEM images were acquired at 200× and 2000× magnification levels using FEI Quanta 200 ESEM operated at 20 Kv voltage and a 10 mm working distance.

SEM images are grey scale images, and it is sometimes difficult to differentiate between the conventional aggregates from crumb rubber aggregates. Therefore, to identify the rubber and conventional aggregate particles, EDS analysis was undertaken at the start of the SEM imaging of all the samples. Figure 5 shows two spectrums taken from the conventional aggregate mortar sample. The first spectrum shows Si and O as the major elements, which are typically the main elements of conventional aggregates. The second spectrum shows a large amount of Ca, O, and Si, along with minor quantities of Al, Fe, and Mg. A large amount of Ca, O, and Si indicates the presence of calcium silicate hydrate gel. Figure 6 shows two spectrums from the adjacent aggregate particles. Both the spectrums show a large amount of carbon (from carbon black and other organic additives) along with a small amount of sulphur (used for vulcanisation), which are the typical elements present in tyre rubber. Figure 8 shows the comparison of all the SEM samples at 200× and 2000× magnification levels.

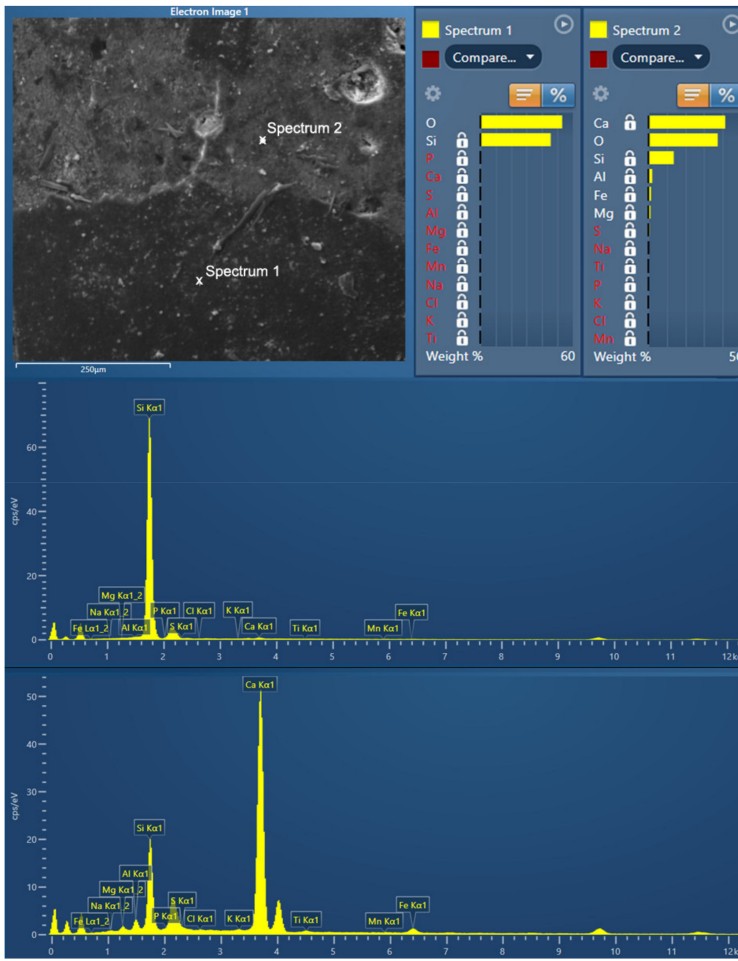

**Figure 5.** EDS analysis for the identification of conventional aggregates in the mortar samples.

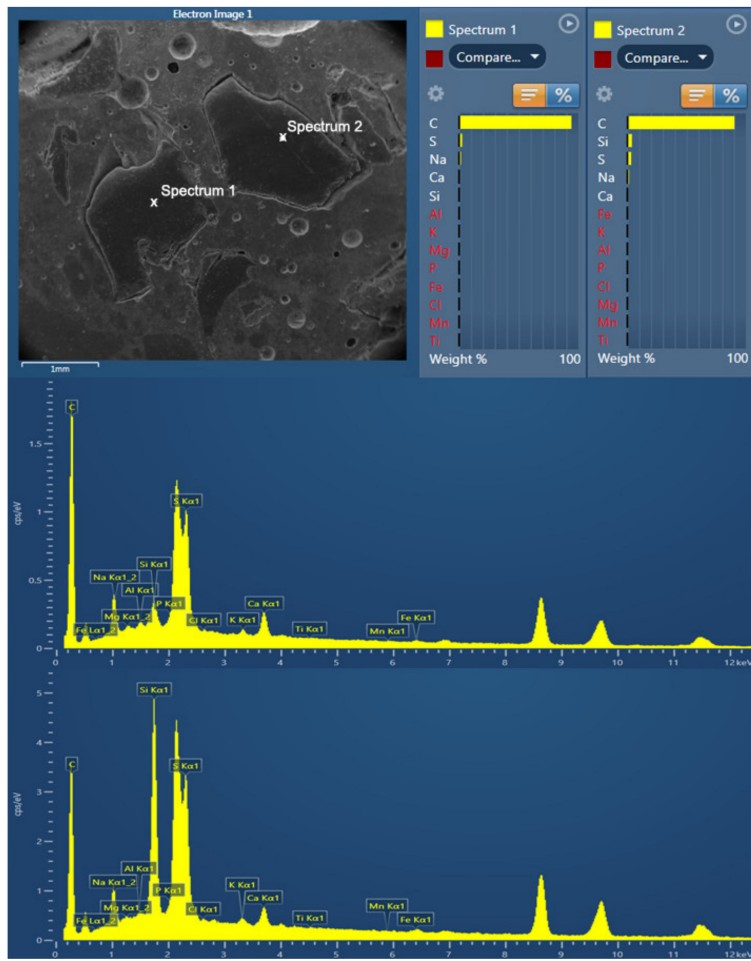

**Figure 6.** EDS analysis for the identification of crumb rubber aggregates in the mortar samples.

## 3. Results and Discussion

The effect of soaking rubber particles in tap water for different durations on the compressive strength of the mortar samples is presented in Figure 7. With the partial replacement of sand by untreated crumb rubber in mix M1, the 7- and 28-day compressive strengths were decreased by 33.8 and 23.1%, respectively, in comparison to that of the control mix. The scanning electron microscopic images of M1 show a large gap between the cement matrix and the rubber particles. Rubber is a soft material and has a very low inherent strength for providing resistance to compressive stresses, thereby acting as a weak void filler material. This combined weakening effect of the ineffective rubber aggregates and the large gap between the cement paste and the rubber particles was most likely responsible for the significant reduction in compressive strength of M1 compared to that of the control mix.

On soaking the rubber particles in plain water for 2 h in M2, the absorbed/adsorbed air on the rubber particles was displaced by water molecules. The SEM images of M2 (Figure 8) show a dark and dense rim of cement paste around the rubber particles, with a seamless interfacial transition zone. This indicates that the air absorbed/adsorbed into the rubber particles repels or acts as a barrier between the cement paste and the rubber particles, which when it gets displaced by water on soaking, helps in the formation of a dense and seamless ITZ. This is a significant improvement for achieving a good cement paste to crumb rubber bond performance, which otherwise required a very long period of water soaking [55] or complex chemical treatments [32,33,56]. Moreover, the darker tone of the rim around the rubber particles also indicates the increased degree of hydration compared to the cement paste outside of the rim. This is most likely because of the additional water

absorbed/adsorbed by the rubber particles, which is available for the hydration of the cement particles that are in close proximity to the rubber particles. Moreover, the mixing procedure of crumb rubber mixes also holds the key to obtaining a dense paste formation at the ITZ between the cement paste and the rubber aggregates. The soaked rubber particles were mixed with the cement powder after straining off the excess water as a first step, which formed a good coating of the cement particles around the rubber aggregates. The increased amount of cement around the rubber particles, in addition to the increased amount of absorbed/adsorbed water available from the crumb rubber particles, was most likely responsible for the formation of a dense rim of hydrated cement paste around the rubber particles of mix M2 compared to M1. This improvement in the bond behaviour and the formation of a dense ITZ around the rubber particles in M2 was also reflected in the 14.8% improvement in its 28-day compressive strength, in comparison to that of mix M1. Compared to the control mix, M2 showed 31.9 and 12.4% reductions in its 7- and 28-day compressive strength results, respectively. Interestingly, the 7-day strength results of M1 and M2 did not show any considerable differences. This could possibly be because the available water molecules absorbed/adsorbed to the rubber particles may not have been depleted at 7 days of curing. However, at later stages, as the internal relative humidity of the cement paste decreases due to the increased degree of hydration, the capillary stresses within the cement paste increased due to the reduction in pore water [57]. This results in the drawing up of this available water by the capillary pores, which not only reduces shrinkage, but is also consumed by the un-hydrated or partially hydrated cement particles [57]. This results in the increase in strength that was evident in the improvement of the 28-day compressive strength of M2, in comparison to that of M1.

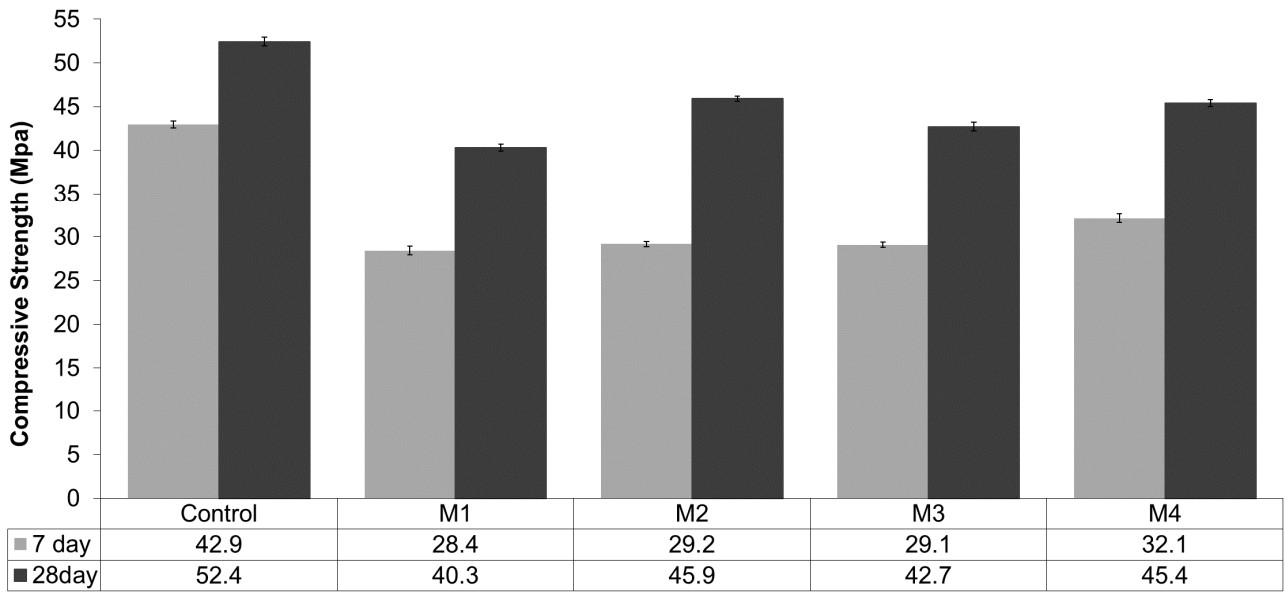

**Figure 7.** Compressive strength results.

When the rubber particles were soaked for a longer duration, i.e., 24 h, a 0.95% increase in water absorption/adsorption was observed in the rubber particles, as shown in Table 4. The SEM image of M3 (Figure 8) at 200× magnification shows a similar darker tone rim around the rubber particles as was visible in M2, indicating the increased degree of hydration. However, the seamless bond between the cement paste and the rubber particles (2000× magnification image) that was visible in the 2-h water-soaked rubber mortar sample (M2) was not evident in the one containing 24 h water-soaked crumb rubber samples (M3). This indicates that the additional absorbed/adsorbed water does not get consumed by the hydration reaction, and thereby forming a thin layer around the rubber particles. This thin layer acts as a barrier in the formation of a seamless bond between the rubber particles and

the cement paste. This was also reflected in a small reduction in the compressive strength of mix M3 in comparison to M2. Compared to the control mix, M3 showed 32.2 and 18.5% reductions in its 7- and 28-day compressive strength results, respectively. However, the increased degree of hydration and a significant reduction in the width of the cavity between the rubber particles and the cement paste resulted in the improvement of compressive strength of M3 in comparison to mix M1.

In mix M4, the crumb rubber particles were soaked in 5% sodium sulphate solution for 2 h. There was a 10% improvement in the 7-day compressive strength results of M4 in comparison to M2; however, no change in the 28-day strength was observed. Compared to the control mix, M4 showed 25.2 and 13.4% reductions in its 7- and 28-day compressive strength results, respectively. The SEM images of M4 show no bond formation between the rubber particle and the cement paste; however, the gap between the paste and the rubber particle was significantly reduced compared to M1. Ettringite crystals were observed in the ITZ, which were further confirmed with the XRD analysis as shown in Figure 9, which shows an increase in the ettringite and portlandite content, and a decrease in the larnite and hatrurite peaks. This indicates that the addition of sodium sulphate not only increased the formation of ettringite, it also accelerated the hydration reaction of larnite and hatrurite, thereby releasing more residual portlandite content. This increase in the ettringite formation and the acceleration of the hydration reaction densified the ITZ, thereby reducing the gap between the rubber particles and the cement paste, which was evident in M1 (Figure 8) containing untreated rubber. The gap formation between the rubber particles and the cement paste was most likely because of the growth of ettringite crystals in the ITZ. Ettringite contains 32 moles of chemically bound water that it draws from its surroundings, thereby reducing the water for hydration of the surrounding cement particles. This sometimes results in a self-desiccation effect, resulting in the shrinkage of the cement paste. This combined effect of the growth of the ettringite crystals in the ITZ and the possible shrinkage of the surrounding gel due to self-desiccation effects [58–61] was most likely the reason behind there being no bond formation between the cement paste and the rubber particles in M4, which was present in M2 containing plain water soaked rubber particles. Overall, this research provides an economical and short duration rubber treatment method that can meet the requirements of the fast-paced construction industry, in comparison to the existing long duration [55] or complex chemical treatment methods [32,33,56] that make it difficult for the construction industry to adopt this waste material.

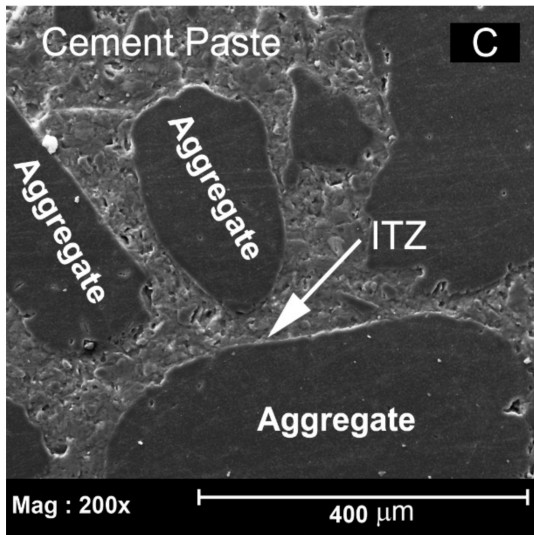 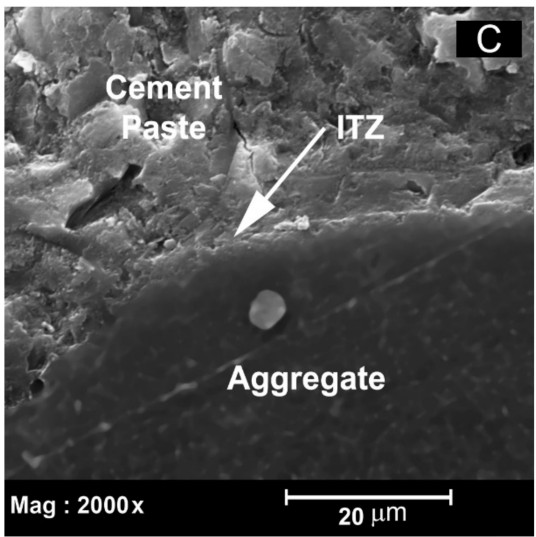

**Figure 8.** *Cont.*

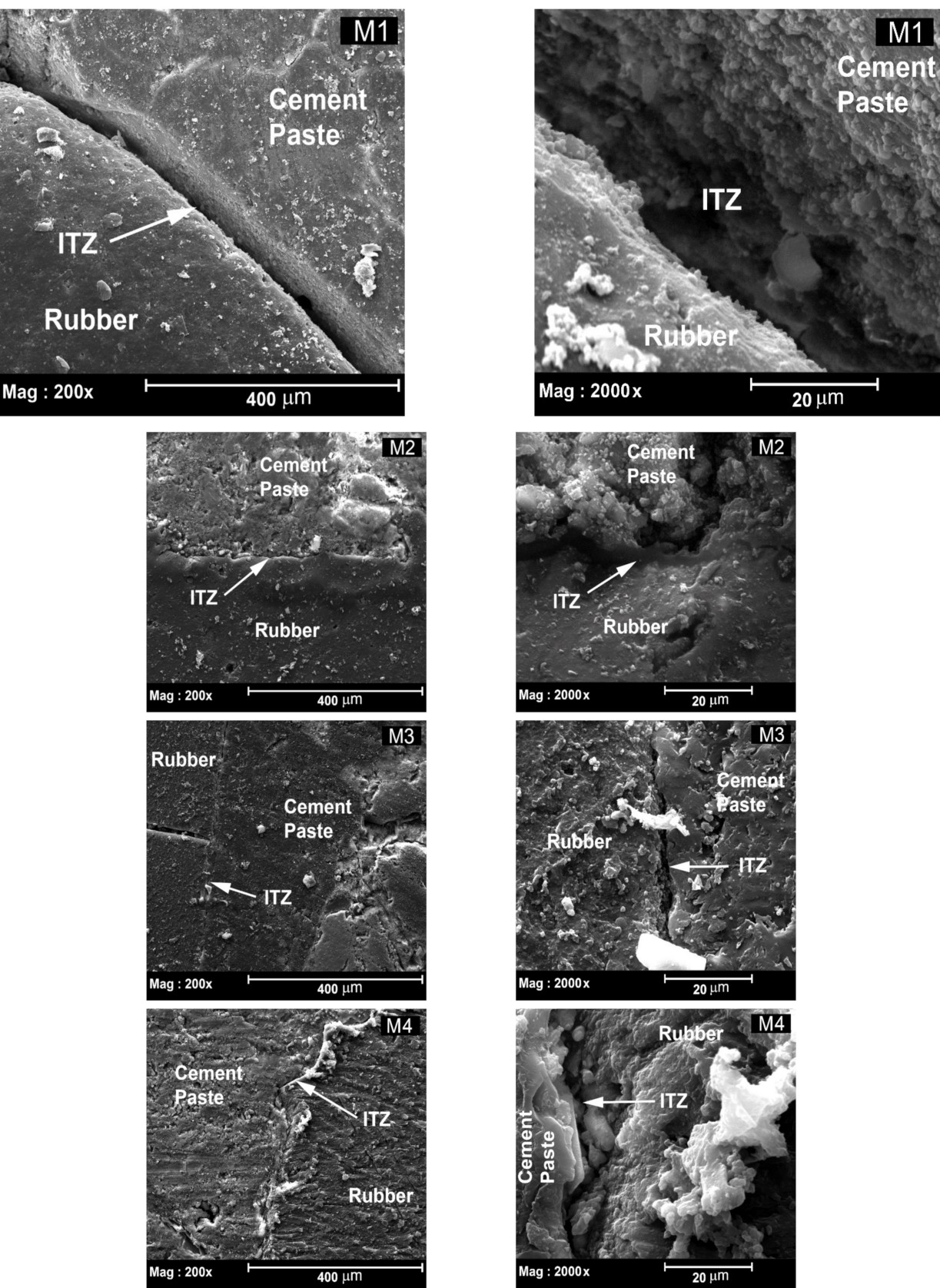

**Figure 8.** SEM images of hardened mortar samples at 28 days of curing.

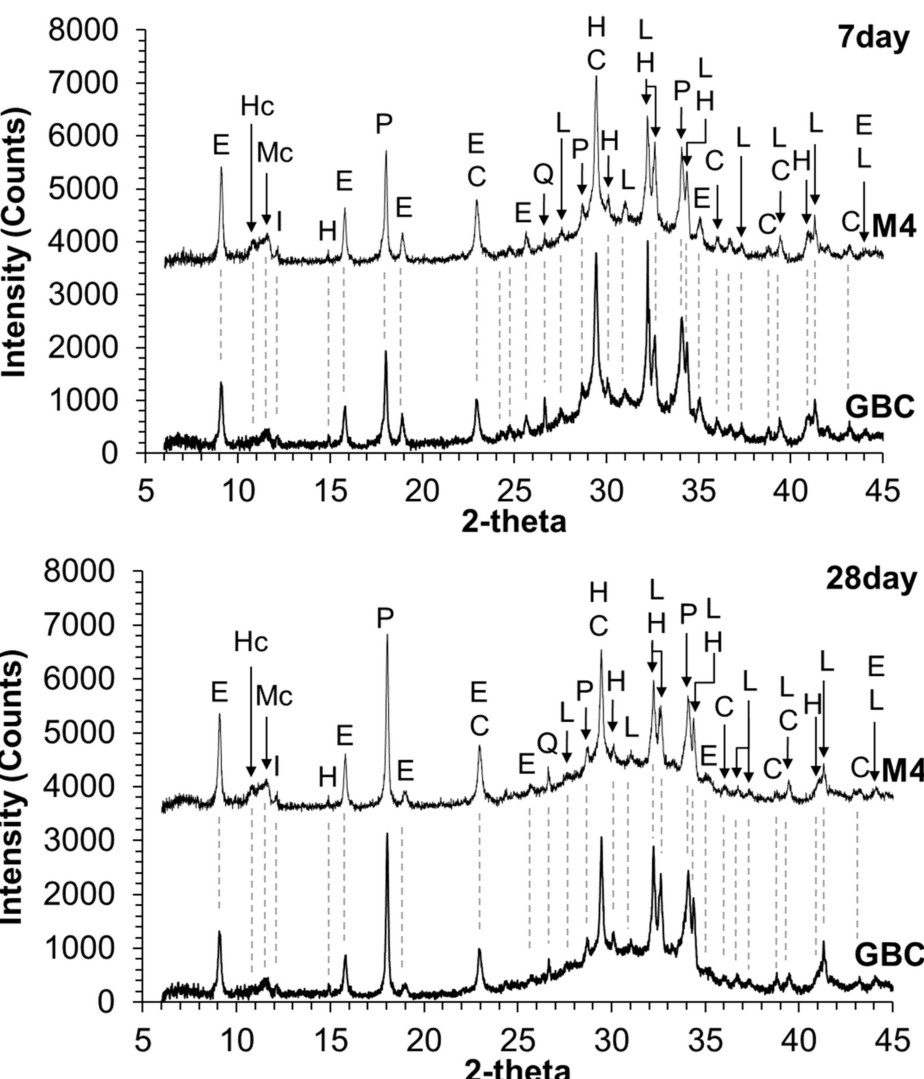

**Figure 9.** X-ray diffractograms of hydrated GBC and M4.

## 4. Conclusions and Recommendations

(C1) Partial replacement of sand by untreated crumb rubber brings about 33.8 and 23.1% reductions in the 7 and 28-day compressive strength results of the crumb rubber concrete, respectively; in comparison to the control mix. The significant reduction in 7- and 28-day strength results of rubber concrete are attributed to the soft nature of the rubber particles and poor bond performance, as evident from a large gap in the interfacial transition zone between the cement paste and the rubber particles. This significant reduction in the compressive strength of rubber concrete is the key issue hindering its application in the construction industry.

(C2) Soaking of crumb rubber for 2 h absorbs/adsorbs ~80% of water compared to 24-h soaked crumb rubber.

(C3) The air adsorbed by the rubber particles and in the pore structure is displaced by the water molecules when soaking in water, which not only helps improve the interfacial transition zone between the rubber particles and the cement paste but also increases the degree of hydration of the cement particles available in close proximity to the rubber particles.

(C4) The soaking of crumb rubber in tap water for 2 h provides much better performance for both the strength and bond properties of the rubber mortar compared to 24 h soaking. It not only improves the interfacial transition zone between the rubber

particles and the cement paste, but also shows an increase in 28-day compressive strength in comparison to 24-h soaked crumb rubber mortar.

(C5) A further improvement in the 7-day strength was achieved with rubber soaked in 5% sodium sulphate solution for 2 h. However, no noticeable improvement in the 28-day compressive strength was observed.

(C6) A significant reduction in soaking time, with the improved performance in strength properties can help in expediting the material handling and construction process of rubber concrete, providing a more practical and economical rubber pre-treatment method for the fast-paced construction industry.

(R1) A recommendation for a future study is to examine reducing the soaking period of crumb rubber by forced submergence and investigate the bond and compressive strength properties.

**Author Contributions:** Conceptualization, R.R.; Data curation, R.R.; Formal analysis, R.R.; Funding acquisition, R.J.G., X.M., Y.Z. and J.E.M.; Investigation, R.R.; Methodology, R.R.; Project administration, R.J.G., X.M., Y.Z. and J.E.M.; Resources, R.R., R.J.G., X.M., Y.Z. and J.E.M.; Software, R.R.; Supervision, R.J.G., X.M., Y.Z. and J.E.M.; Writing—original draft, R.R. and O.Y.; Writing—review & editing, R.R., O.Y., R.J.G., X.M., Y.Z. and J.E.M. All authors have read and agreed to the published version of the manuscript.

**Funding:** The authors would like to acknowledge the funding provided by the Australian Research Council (ARC-LP160100298) and industry partners for this project.

**Acknowledgments:** The authors would like to acknowledge the funding provided by the Australian Research Council (ARC-LP160100298) and industry partners for this project. The industry partners are: Tyrecycle Pty Ltd., Tyre Stewardship Australia, ResourceCo Pty Ltd., FMG Engineering Pty Ltd., and Ancon Beton. The material donations by ResourceCo Pty. Ltd., Adelaide Brighton Cement Pty. Ltd., and Tyrecycle Pty. Ltd., are greatly appreciated. The authors also would like to thank the support of lab staff members in University of South Australia. The authors fully acknowledge the support provided by the civil lab, Rheology and material characterisation, Microscopy & microanalysis, and the X-ray facilities at RMIT University.

**Conflicts of Interest:** The authors declare no conflict of interest.

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
