# Peer review of "Practical Rubber Pre-Treatment Approch for Concrete Use—An Experimental Study"

_jcs, doi:10.3390/jcs5060143_

Round 1
Reviewer 1 Report
This paper investigates the effect of treating crumb rubber particles on the compressive strength of concrete. The following are some comments to the authors to consider in the revised manuscript:
- In the Experimental Program section, center the position of Figure 1.
- In the Experimental Program section, add description on how the rubber particles were prepared to the right size. Were they randomly cut from a tyre? How they were cut? Discuss how this affects the overall process of preparing the concrete mix regarding time and cost.
- In the results and discussion section, the authors based all of their comparison and analysis on comparing M2, M3 and M4 properties to M1 results showing an increase in the 28-day compressive strength and not much an increase in the 7-day strength. However, the comparison should be based on the control mix properties, which shows that adding rubber to the concrete decreased both the 7-day and 28-day compressive strengths.
- The authors need to clarify and mention in the discussion and conclusion section that adding rubber particles did not improve the compressive strength of concrete and discuss what the reasons are and how this can limit the use of this type of concrete.
Author Response
Dear Reviewer,
We thank you for the valuable and constructive feedback. We have revised the manuscript to address your comments. the responses to your comments are as below:
Comment 1: In the Experimental Program section, center the position of Figure 1.
Response 1: We have centered the position of Figure 1 to suit the reviewer’s comment
Comment 2: In the Experimental Program section, add description on how the rubber particles were prepared to the right size. Were they randomly cut from a tyre? How they were cut? Discuss how this affects the overall process of preparing the concrete mix regarding time and cost.
Response 2: Please note that the crumb rubber was sourced from a tyre recycling company Tyrecycle (https://www.tyrecycle.com.au/). They have industrial scale tyre shredding facility that produces crumb rubber from waste tyres. We have revised Section 2 (Experimental program) to incorporate this information.
Comment 3: In the results and discussion section, the authors based all of their comparison and analysis on comparing M2, M3 and M4 properties to M1 results showing an increase in the 28-day compressive strength and not much an increase in the 7-day strength. However, the comparison should be based on the control mix properties, which shows that adding rubber to the concrete decreased both the 7-day and 28-day compressive strengths.
Response 3: We have revised the manuscript to add the comparison of the treated rubber mixes with the control mix to suit the reviewer’s comment. The comparison with mix M1 was to highlight the benefit of rubber treatment over untreated rubber.
Comment 4: The authors need to clarify and mention in the discussion and conclusion section that adding rubber particles did not improve the compressive strength of concrete and discuss what the reasons are and how this can limit the use of this type of concrete.
Response 4: We have added another conclusion point “C1” about the negative effect of the addition of crumb rubber. The first paragraph of the results and discussion section talks about the negative effect of the addition of crumb rubber to concrete along with the reasons. The subsequent sections were to improve these properties by using different rubber treatment methods.

Reviewer 2 Report
In the paper is presented a serie of information on pre-treatment of rubber waste used in the manufacture of concrete.
From the analysis of the information presented in the article, I found the following:
- The paper presents a series of results that may be of interest to the scientific community:
- The first part of the summary should be deleted because it does not present the results of the research carried out but some generalities related to rubber waste;
-From the introductory part, the first paragraph must be deleted;
- There is a distribution of rubber and sand particles, but no method is used to determine it;
- A more in-depth analysis of the characteristics of the rubber particles must be presented. A presentation of the condition of the particle surface is required. Thus, they can be particles with more rough or less rough surfaces. This roughness of the rubber particles can influence the adhesion with the cement that enters the concrete structure;
- The part of the research methodology must be completed with justifications regarding the decision to choose the composition of the test samples used in the research;
- Macroscopic images with the test samples used in the research must also be presented;
- The discussion part must be completed in such a way as to highlight the novelty brought by the research in relation to other research in the field;
- The conclusions should include a series of information on the practical possibility of using the research presented, as well as future research directions.
Author Response
Dear Reviewer,
We thank you for your valuable and constructive feedback. We have revised the manuscript to address your comments.
Comment 1: The paper presents a series of results that may be of interest to the scientific community:
Response 1: We thank the reviewer for his/her positive comment
Comment 2: The first part of the summary should be deleted because it does not present the results of the research carried out but some generalities related to rubber waste.
Response 2: We have deleted the first two lines from the Abstract (summary) section that were general statements about the crumb rubber waste, to suit the reviewer’s comment
Comment 3: From the introductory part, the first paragraph must be deleted
Response 3: We have deleted the the first paragraph from the introduction section to suit the reviewer’s comment
Comment 4: There is a distribution of rubber and sand particles, but no method is used to determine it
Response 4: Please note that we carried out sieve analysis to ascertain the particle distribution of sand and crumb rubber. It is provided in “Section 2: Experimental Programme” (last two lines)
Comment 5: A more in-depth analysis of the characteristics of the rubber particles must be presented. A presentation of the condition of the particle surface is required. Thus, they can be particles with more rough or less rough surfaces. This roughness of the rubber particles can influence the adhesion with the cement that enters the concrete structure
Response 5: We have collected two random samples from the same batch of crumb rubber and collected SEM images from those samples, to address the reviewer’s comment. Overall, the texture of the all crumb rubber particles used in this study have similar rough surface.
Comment 6: The part of the research methodology must be completed with justifications regarding the decision to choose the composition of the test samples used in the research
Response 6: We have added a paragraph on page number 5 describing the reasoning behind choosing the 20% composition of crumb rubber in the mix designs
Comment 7: Macroscopic images with the test samples used in the research must also be presented
Response 7: Unfortunately, we didn’t take any pictures at the time of sample testing.
Comment 8: The discussion part must be completed in such a way as to highlight the novelty brought by the research in relation to other research in the field
Response 8: We have highlighted the novelty of this work by comparing the benefits of the current research relative to the existing published work on page 15 and 19.
Comment 9: The conclusions should include a series of information on the practical possibility of using the research presented, as well as future research directions.
Response 8: Conclusion “C6” describes the benefits of the current experimental work and its practical feasibility of adoption by the fast paced construction industry. Recommendation “R1” describes the recommendation for the future study to reduce the soaking period of crumb rubber by forced submergence and investigate its bond and compressive strength properties.
Round 2
Reviewer 2 Report
It should also be noted that these aging conditions correspond to a duration of 15 years of use of the part of a car.